# Fault-Tolerant Model Predictive Control Algorithm for Path Tracking of Autonomous Vehicle

**DOI:** 10.3390/s20154245

**Published:** 2020-07-30

**Authors:** Keke Geng, Nikolai Alexandrovich Chulin, Ziwei Wang

**Affiliations:** 1School of Mechanical Engineering, Southeast University, Nanjing 211189, China; 213162730@seu.edu.cn; 2School of Automation Systems, Moscow Bauman State Technical University, Moscow 109807, Russia; nchulin@yandex.ru

**Keywords:** autonomous vehicle, model predictive control, path tracking control, fault detection and isolation

## Abstract

The fault detection and isolation are very important for the driving safety of autonomous vehicles. At present, scholars have conducted extensive research on model-based fault detection and isolation algorithms in vehicle systems, but few of them have been applied for path tracking control. This paper determines the conditions for model establishment of a single-track 3-DOF vehicle dynamics model and then performs Taylor expansion for modeling linearization. On the basis of that, a novel fault-tolerant model predictive control algorithm (FTMPC) is proposed for robust path tracking control of autonomous vehicle. First, the linear time-varying model predictive control algorithm for lateral motion control of vehicle is designed by constructing the objective function and considering the front wheel declination and dynamic constraint of tire cornering. Then, the motion state information obtained by multi-sensory perception systems of vision, GPS, and LIDAR is fused by using an improved weighted fusion algorithm based on the output error variance. A novel fault signal detection algorithm based on Kalman filtering and Chi-square detector is also designed in our work. The output of the fault signal detector is a fault detection matrix. Finally, the fault signals are isolated by multiplication of signal matrix, fault detection matrix, and weight matrix in the process of data fusion. The effectiveness of the proposed method is validated with simulation experiment of lane changing path tracking control. The comparative analysis of simulation results shows that the proposed method can achieve the expected fault-tolerant performance and much better path tracking control performance in case of sensor failure.

## 1. Introduction

Fault signal detection and isolation, as well as fault-tolerant control systems, are important contents in the research field of autonomous vehicle and prerequisites for ensuring the driving safety in complex traffic scenarios. The failure of autonomous vehicles mainly occurs in the process of sensing information acquisition and motion state transmission. Fault detection and isolation algorithms are widely used in various unmanned systems, such as ground autonomous vehicles [1], underwater robots [2], and autonomous helicopters [3]. For autonomous vehicle systems, fault signal detection and isolation algorithms play an important role in autonomous environment perception, decision making, and motion control. In order to effectively detect and isolate fault signals and perform stable motion control, many schemes and technologies have been proposed, which can be divided into: the nonlinear algorithms [4] and the linear algorithms [5,6]. With the continuous development of autonomous vehicle technology, the types and number of on-board sensors also continue to increase, and the fault sensor signals have become the main reason for vehicle failure [7,8,9].

Sensor failure detection and isolation based on model information is a commonly used method [10]. Marzat J. and Avram R.C. et al. proposed sensor fault detection and isolation algorithms using control model information [11] and sliding mode observer [12], respectively. The evaluation and performance of model-based fault detection and isolation algorithms always depend on the accuracy of the system model used. Because the vehicle is a highly coupled and complex nonlinear system, it is impossible to obtain some vehicle parameters accurately, and the vehicle system model established always has uncertainty. In order to better detect faults and isolate fault signals at the site, some methods and strategies that are not model-based are proposed. Methods such as fuzzy logic [13,14], neural network [15,16], and Kalman filter [17] are used to estimate uncertain parameters in nonlinear systems. The performance of model-based fault detection and isolation methods rely on accurate linear system modeling. For the nonlinear systems [18], satisfactory results cannot be obtained by using these methods. However, the model-based methods that require less calculation and real-time performance are also better. Therefore, those model-based methods are still widely used in solving real engineering problems.

Recently, model-based fault detection and isolation algorithms for vehicle systems have been extensively studied. In particular, Chamseddine [19] used a sliding mode observer based on a quarter car model to detect sensor failures in vehicle systems. Although fault detection algorithms are robust to interferences, additional sensors such as displacement sensors are still used to replace the commonly used sensor configurations in the commercial vehicles [20]. In References [21,22], the parity check space method, which is typical model-based method, is proposed for fault detection. The applicability and stability of the model-based fault detection and isolation algorithms are usually poor because of the model uncertainty, interference, and sensor noise. In addition, few scholars have studied fault signal detection and isolation algorithms for robust path tracking control autonomous vehicles.

In order to overcome these limitations, this paper proposes a robust fault-tolerant model predictive control algorithm for path tracking of autonomous vehicle. The nonlinear single-track dynamic vehicle model is established as the research object and the linearization is carried out by using Taylor expansion. The model predictive control algorithm is designed for lateral path tracking control of autonomous vehicle. A fault signal detection and isolation algorithm is proposed and its implementation process can be described as follows: first, based on the output error covariance and weighted data fusion method, the optimal motion state information of the autonomous vehicle is obtained. Then, we designed a fault signal detector, composed of a main Kalman filter, three sub-Kalman filters, two state Chi-square detectors, and residuals Chi-square detectors, with fault signal detection matrix as output. Finally, the fault signal matrix is constructed as a diagonal matrix, which will be multiplied by the sensor signal matrix and the weight coefficient matrix to realize fault signal isolation. The single lane changing path tracking control simulation results confirmed the effectiveness of the proposed method in this paper.

The flowchart of the fault-tolerant model predictive control algorithm designed in this paper is shown in Figure 1. Vehicle motion status information can usually be obtained by GPS combined navigation system, visual odometer, and LIDAR SLAM. These sensors and algorithms together form an on-board perception system. 

The rest of this paper is organized as follows. In Section 2, the vehicle dynamic model is established and model linearization process is described. In Section 3, the constraints and objective functions are constructed, and the path tracking control algorithm based on model predictive control is designed. In Section 3, the multi-sensor information fusion algorithm, fault signal detection algorithm, and isolation algorithm are described. Section 4 presents experimental verification using lane changing path tracking scenario. Conclusions are given in Section 5.

## 2. Modeling and Problem Linearization

The impacts of the vehicle suspension characteristics are relatively small in relation to the research content of vehicle motion control. In this work, the vehicle-tire model is selected, which means no in-depth research on the characteristics of vehicle suspension. At the same time, the dynamic model established in this paper is mainly used to design the predictive model in the model predictive controller. It is required to simplify as much as possible on the basis of more accurately describing the vehicle’s dynamic characteristics and reducing the amount of calculation. The following idealized assumptions are first proposed when performing dynamic modeling: (1) Ignoring road fluctuations and assuming that the vehicle is always driving on a flat road without vertical motion; (2) Ignoring suspension motion and the effect of the suspension structure on the coupling relationship; (3) The load movement of the front and rear axles is not considered, and the left and right transfer of the load is ignored; (4) Only the tire cornering characteristics are considered, and the vertical and horizontal coupling relationships are ignored; (5) Mechanical effects of steering system are also ignored. In this paper, a 3-degree-of-freedom single-track vehicle dynamics model is constructed, including longitudinal motion, lateral motion, and yaw (see Figure 2).

The longitudinal force, lateral force, and yaw motions of the vehicle can be written as:(1)$$\left\{ \begin{array}{l} m\left(\ddot{x}-\dot{y}\dot{\varphi }\right)=\sum F_{x}=2F_{lf}\cos\delta_{f}-2F_{cf}\sin\delta_{f}+2F_{lr}\\ m\left(\ddot{y}+\dot{x}\dot{\varphi }\right)=\sum F_{y}=2F_{lf}\sin\delta_{f}-2F_{cf}\cos\delta_{f}+2F_{cr}\\ I_{z}\ddot{\varphi }=\sum M_{z}=2L_{f}\left(F_{lf}\sin\delta_{f}+F_{cf}\cos\delta_{f}\right)-2L_{r}F_{cr} \end{array}\right.$$
where *m* is the vehicle mass; φ is the yaw angle; *x* and *y* are the longitudinal and lateral position, respectively; $\delta_{f}$ is the front wheel rotation angle; $I_{z}$ is the *z*-axis moment of inertia; $F_{x}$ is the total longitudinal force on the vehicle; $F_{y}$ is total lateral force on the vehicle; $M_{z}$ is the total yaw moment on the vehicle; $F_{cf}$, $F_{cr}$ are the lateral forces on the front and rear tires of the vehicle, and are related to the corner stiffness and corner angle of vehicle tires; $F_{lf}$, $F_{lr}$ are longitudinal forces on the front and rear tires of the vehicle, which are related to the longitudinal stiffness and slip rate of the tire; $F_{xf}$, $F_{xr}$ are the forces on the front and rear tires in the *x* direction; $F_{yf}$, $F_{yr}$ are the forces on the front and rear tires in the *y* direction; $L_{f}$ and $L_{r}$ are the distances from the front and rear axis to the center of mass. 

According to Equation (1), the vehicle dynamics model involves vehicle tire forces. The longitudinal and lateral forces are related to vertical load, road friction coefficient, slip rate, and tire corner angle:(2)$$\left\{ \begin{array}{l} F_{l}=f_{l}\left(F_{z},s,\mu ,\alpha \right)\\ F_{c}=f_{c}\left(F_{z},s,\mu ,\alpha \right)\\ s=\left\{ \begin{array}{l} rw_{t}/v-1,\;\left(v>rw_{t},\;v\neq 0\right)\\ 1-rw_{t}/v,\;\left(v<rw_{t},\;w_{t}\neq 0\right) \end{array}\right.\\ \alpha =\tan^{-1}\left(v_{c}/v_{l}\right) \end{array}\right.$$
where $F_{z}$ is the vertical load; *s* is the slip rate; *μ* is the road surface adhesion coefficient; *α* is the tire cornering angle; $w_{t}$ is the wheel speed; *r* is the wheel radius; $v_{l}$ is the longitudinal speed; and $v_{c}$ is the lateral speed, which can be expressed by $v_{x}$ and $v_{y}$:(3)$$\left\{ \begin{array}{l} v_{l}=v_{y}\sin\delta_{y}+v_{x}\cos\delta_{y}\\ v_{c}=v_{y}\cos\delta_{y}-v_{x}\sin\delta_{y} \end{array}\right.$$

Generally, the tire speed of a vehicle is difficult to obtain directly, which can be obtained by calculating the vehicle speed:(4)$$\left\{ \begin{array}{l} v_{yf}=\dot{y}+L_{f}\dot{\varphi }\\ v_{yr}=\dot{y}-L_{r}\dot{\varphi }\\ v_{xf}=\dot{x}\\ v_{xr}=\dot{x} \end{array}\right.$$

When constructing the vehicle dynamics model, the front and rear axle load movements have been ignored. Therefore, the vertical load on the front and rear wheels of the vehicle can be calculated as:(5)$$\left\{ \begin{array}{l} F_{zf}=L_{r}mg/\left(2L_{f}+2L_{r}\right)\\ F_{zr}=L_{f}mg/\left(2L_{f}+2L_{r}\right) \end{array}\right.$$

Generally, a vehicle in a stable driving state has a small variation angle and a slip rate. According to the Semi-Empirical Tire-Model [23], we know that the tire dynamics, including the tire longitudinal force, the tire lateral force, and the tire aligning torque, have obvious nonlinear characteristics, but the simulation results in References [24,25] show that the tire forces can be approximated by a linear equation when the longitudinal slip rate and the tire variation angle change in a small range. In addition, there are a large number of trigonometric functions in the vehicle dynamics model. Since each angle involved in the dynamics model is in a small angle interval, each trigonometric function can satisfy the following approximate conditions: cosθ≈1, sinθ = 0, tanθ = θ. After introducing the corner stiffness, corner angle, longitudinal stiffness, and slip rate, the tire force of the vehicle can be expressed as:(6)$$\left\{ \begin{array}{l} F_{lf}=C_{lf}s_{f}\\ F_{lr}=C_{lr}s_{r}\\ F_{cf}=C_{cf}\left[\delta_{f}-\left(\dot{y}+L_{f}\dot{\varphi }\right)/\dot{x}\right]\\ F_{cr}=C_{cr}\left(L_{r}\dot{\varphi }-\dot{y}\right)/\dot{x} \end{array}\right.$$
where $C_{cf}$, $C_{cr}$ are the lateral stiffness of the front and rear tires; $C_{lf}$, $C_{lr}$ are the longitudinal stiffness of the front and rear tires; $S_{f}$, $S_{r}$ are the slip ratio of the front and rear tires.

Nonlinear vehicle dynamics model can be written as:(7)$$\left\{ \begin{array}{l} \ddot{x}=\dot{y}\dot{\varphi }+\left(2/m\right)\left\{C_{lf}s_{f}-C_{cf}\left[\delta_{f}-\left(\dot{y}+L_{f}\dot{\varphi }\right)/\dot{x}\right]\delta_{f}+C_{lr}s_{r}\right\}\\ \ddot{y}=-\dot{x}\dot{\varphi }+\left(2/m\right)\left\{C_{lf}s_{f}\delta_{f}+C_{cf}\left[\delta_{f}-\left(\dot{y}+L_{f}\dot{\varphi }\right)/\dot{x}\right]-C_{cr}\left(\dot{y}-L_{r}\dot{\varphi }\right)/\dot{x}\right\}\\ \dot{\varphi }=\dot{\varphi }\\ \ddot{\varphi }=\left(2L_{f}/I_{z}\right)\left\{C_{lf}s_{f}\delta_{f}+C_{cf}\left[\delta_{f}-\left(\dot{y}+L_{f}\dot{\varphi }\right)/\dot{x}\right]\right\}+\left(2L_{r}/I_{z}\right)C_{cr}\left(\dot{y}-L_{r}\dot{\varphi }\right)/\dot{x}\\ \dot{Y}=\dot{x}\sin\varphi +\dot{y}\cos\varphi \\ \dot{X}=\dot{x}\cos\varphi -\dot{y}\sin\varphi \end{array}\right.$$

For the convenience, $\xi_{dyn}={\left[\dot{x},\dot{y},\varphi ,\dot{\varphi },X,Y\right]}^{T}$ are system state quantities and $u_{dyn}=\delta_{f}$ is the system control quantity.

### Linearization of Vehicle Dynamics Model

For autonomous vehicle, the lateral motion control is to control the front wheel rotation angle, and then realize path tracking. Therefore, this paper selects path tracking as the ultimate goal of autonomous vehicle lateral control, and the tracking accuracy as the main indicator to measure the performance of the control system. Model predictive control can be divided into linear time-varying model predictive control (LMPC) [26] and nonlinear model predictive control (NMPC) [27]. Compared with NMPC, the LMPC uses the linear predictive model and has better real-time performance, which is a very important character for the motion control of autonomous vehicles. Thus, the LMPC is used in this work.

The vehicle model established in this work is a nonlinear model, which needs to be linearized. The state quantity and control quantity of the system satisfy the following relationship:(8)$${\dot{\xi }}_{r}=f\left(\xi_{r},u_{r}\right)$$

Perform Taylor expansion at $\left(\xi_{r},u_{r}\right)$, retain the first-order terms, and ignore the higher-order terms, we get:(9)$$\dot{\xi }=f\left(\xi_{r},u_{r}\right)+{\frac{\partial f}{\partial f\xi }|}_{\begin{array}{c} \xi =\xi_{r}\\ u=u_{r} \end{array}}\left(\xi -\xi_{r}\right)+{\frac{\partial f}{\partial u}|}_{\begin{array}{c} \xi =\xi_{r}\\ u=u_{r} \end{array}}\left(u-u_{r}\right)$$

The formula can be transformed into:(10)$$\dot{\xi }=f\left(\xi_{r},u_{r}\right)+J_{f}\left(\xi \right)\left(\xi -\xi_{r}\right)+J_{f}\left(u\right)\left(u-u_{r}\right)$$
where $J_{f}\left(\xi \right)$ and $J_{f}\left(u\right)$ are the Jacobian matrixes of *f* relative to ξ and *u*, respectively.

Subtract the formula to get:(11)$$\dot{\tilde{\xi }}=A\left(t\right)\tilde{\xi }+B\left(t\right)\tilde{u}$$

The linearized system equation can be written as:
(12)$$\begin{array}{l} \dot{\xi }=A\left(t\right)\xi \left(t\right)+B\left(t\right)u\left(t\right)\\ y=C\xi \left(t\right) \end{array}$$
where A(t)=∂f/∂ξ, B(t)=∂f/∂u, $C={\left(0,0,0,0,1,0\right)}^{T}$, and:(13)$$A\left(t\right)=\left[\begin{array}{cccccc} -\frac{2C_{cf}\delta_{f,t-1}\left({\dot{y}}_{t}+L_{f}{\dot{\varphi }}_{t}\right)}{m{\dot{x}}_{t}^{2}} & {\dot{\varphi }}_{t}+\frac{2C_{cf}\delta_{f,t-1}}{m{\dot{x}}_{t}} & 0 & {\dot{y}}_{t}+\frac{2C_{cf}\delta_{f,t-1}L_{f}}{m{\dot{x}}_{t}} & 0 & 0\\ \frac{2\left[C_{cf}\left({\dot{y}}_{t}+L_{f}{\dot{\varphi }}_{t}\right)-C_{cr}\left(L_{r}{\dot{\varphi }}_{t}-{\dot{y}}_{t}\right)\right]}{m{\dot{x}}_{t}^{2}}-{\dot{\varphi }}_{t} & -\frac{2\left(C_{cf}+C_{cr}\right)}{m{\dot{x}}_{t}} & 0 & -{\dot{x}}_{t}+\frac{2\left(C_{cr}L_{r}-C_{cf}L_{f}\right)}{m{\dot{x}}_{t}} & 0 & 0\\ 0 & 0 & 0 & 1 & 0 & 0\\ \frac{2\left[C_{cf}L_{f}\left({\dot{y}}_{t}+L_{f}{\dot{\varphi }}_{t}\right)+C_{cr}L_{r}\left(L_{r}{\dot{\varphi }}_{t}-{\dot{y}}_{t}\right)\right]}{I_{z}{\dot{x}}_{t}^{2}} & \frac{2\left(C_{cr}L_{r}-C_{cf}L_{f}\right)}{I_{z}{\dot{x}}_{t}} & 0 & \frac{-2\left(C_{cf}L_{f}^{2}+C_{cr}L_{r}^{2}\right)}{I_{z}{\dot{x}}_{t}} & 0 & 0\\ \cos\left(\varphi_{t}\right) & -\sin\left(\varphi_{t}\right) & -{\dot{x}}_{t}\sin\left(\varphi_{t}\right)-{\dot{y}}_{t}\cos\left(\varphi_{t}\right) & 0 & 0 & 0\\ \sin\left(\varphi_{t}\right) & \cos\left(\varphi_{t}\right) & {\dot{x}}_{t}\cos\left(\varphi_{t}\right)-{\dot{y}}_{t}\sin\left(\varphi_{t}\right) & 0 & 0 & 0 \end{array}\right]$$
(14)$$B\left(t\right)=\left[\begin{array}{cccccc} -\frac{2C_{cf}}{m}\left(2\delta_{f,t-1}-\frac{{\dot{y}}_{t}+L_{f}{\dot{\varphi }}_{t}}{{\dot{x}}_{t}}\right) & \frac{2\left(C_{lf}s_{f}+C_{cf}\right)}{m} & 0 & \frac{2L_{f}\left(C_{lf}s_{f}+C_{cf}\right)}{m} & 0 & 0 \end{array}\right]$$

Discrete the above formula using the first-order difference quotient method to obtain the discrete state space equation:(15)$$\left\{ \begin{array}{l} \xi \left(k+1\right)=A\left(k\right)\xi \left(k\right)+B\left(k\right)u\left(k\right)\\ \zeta \left(k\right)=C\xi \left(k\right) \end{array}\right.$$
where A(k)=I+TA(t), B(k)=I+TB(t), and T is sampling time.

After introducing the incremental model, the state-space equation can be written as:(16)$$\left\{ \begin{array}{l} \Delta \xi \left(k+1\right)=A\left(k\right)\Delta \xi \left(k\right)+B\left(k\right)\Delta u\left(k\right)\\ \zeta \left(k\right)=C\Delta \xi \left(k\right) \end{array}\right.$$

## 3. Model Predictive Controller for Vehicle Lateral Motion Control

### 3.1. Construct the Objective Function

This paper uses the following objective function:(17)$$J\left(k\right)=\sum_{i=1}^{N_{p}}{\left\|\Delta \eta \left(k+i|k\right)\right\|}_{Q}^{2}+\sum_{i=1}^{N_{c}-1}{\left\|\Delta u\left(k+i|k\right)\right\|}_{R}^{2}+\rho \varepsilon^{2}$$
where ρ is the weight coefficient; ε is the relaxation factor; $N_{p}$ is the prediction time domain and $N_{c}$ is the control time domain; Q is the state weighting matrix; R is the control weighting matrix; ∆η(k+i|k) is output deviation; and ∆u(k+i|k) is control deviation.

Since the vehicle dynamic model is used and the number of constraints is increased, in order to avoid the occurrence of no optimal solution, a relaxation factor ε is added to the objective function.

### 3.2. Construct the Constraints

The most prominent feature of the model predictive control is that it can easily handle the multi-constraint problem. In order to ensure that the reference path can be smoothly tracked, this paper uses the front wheel declination constraint, the front wheel declination incremental constraint, and the tire lateral angle dynamic constraints.

#### 3.2.1. Front Wheel Declination and Its Incremental Constraints

Restrictions on the front wheel deflection angle and front wheel deflection angle of the vehicle can be set according to the actual physical parameters of the vehicle. The control quantity constraint expression is:(18)$$u_{\min}\left(k+t\right)<u\left(k+t\right)<u_{\max}\left(k+t\right),\;k=0,1,\cdots ,N-1$$

The expression of the incremental constraints is:(19)$$\Delta u_{\min}\left(k+t\right)<\Delta u\left(k+t\right)<\Delta u_{\max}\left(k+t\right),\;k=0,1,\cdots ,N-1$$

In the objective function and constraints, the optimized variable is the control quantity increment in the control time domain. Therefore, the control variable must first be converted into the matrix form of Δu. 

The relationship between the control increment and the control quantity can be obtained:(20)$$\left\{ \begin{array}{l} u\left(t+1\right)=\Delta u\left(t\right)+u\left(t\right)\\ u\left(t+2\right)=\Delta u\left(t+1\right)+\Delta u\left(t\right)+u\left(t\right)\\ \cdots \\ u\left(t+N\right)=\Delta u\left(t+N-1\right)+\Delta u\left(t+N-2\right)+\cdots +\Delta u\left(t\right)+u\left(t\right) \end{array}\right.$$

Convert Equation (18) to $U_{\min}\leq A\Delta U+U_{t}\leq U_{\max}$, where:(21)$$A=\left[\begin{array}{cccc} 1 & 0 & \cdots & 0\\ 1 & 1 & \cdots & 0\\ \vdots & \vdots & & \vdots \\ 1 & 1 & \cdots & 1 \end{array}\right],\;\Delta U={\left[\begin{array}{c} \Delta u_{\min}\left(t+1\right)\\ \Delta u_{\min}\left(t+2\right)\\ \vdots \\ \Delta u_{\min}\left(t+N\right) \end{array}\right]}_{N\times 1},\;U_{t}={\left[\begin{array}{c} u\left(t\right)\\ u\left(t\right)\\ \vdots \\ u\left(t\right) \end{array}\right]}_{N\times 1}$$

The constraints of control quantity and control increment ensure that the control output generated by the model prediction controller is physically achievable, but for driving safety and comfort, the dynamic constraints of the vehicle also need to be introduced.

#### 3.2.2. Dynamic Constraint of Tire Cornering

The vehicle sideslip due to wet or slippery roads may cause various accidents. Therefore, it is particularly important to increase vehicle dynamics constraints and reduce the possibility of vehicle sideslip.

The sideslip of the vehicle is closely related to the tire slip angle. When the vehicle runs straight on a horizontal road, the tire slip angle α=0; when the tire is elastically deformed by lateral force without lateral slip, $\alpha \;\leq \alpha_{max}$; when the tire is subjected to excessive lateral force, the vehicle slips, $\alpha \;>\;\alpha_{max}$. It can be concluded that the slip angle of the vehicle tire directly reflects whether the vehicle is slipping, and limiting the tire slip angle limits the occurrence of sideslip.

Since the established vehicle dynamics state-space equation does not take the tire slip angle as a state quantity and cannot directly constrain the tire slip angle, this paper needs to find the relationship between the tire slip angle *α* and the state quantity ξ*(k,t)*. The relationship is to restrain the tire slip angle by imposing a specific relationship constraint on the state quantity.

By Equations (2)–(4), the available tire front and rear wheel angles are:(22)$$\left\{ \begin{array}{l} \alpha_{f}=\left(\dot{y}+L_{f}\dot{\varphi }\right)/\dot{x}-\delta_{f}\\ \alpha_{r}=\left(\dot{y}-L_{r}\dot{\varphi }\right)/\dot{x} \end{array}\right.$$

Using $\xi_{dyn}$ as the state quantity and $u_{dyn}$ as the control quantity, linearize the above formula to obtain:(23)$$\left\{ \begin{array}{l} \alpha =E\xi \left(k,t\right)+Fu_{dyn}\left(k,t\right)\\ E=\left[\begin{array}{cccccc} \frac{1}{\dot{x}} & -\frac{\dot{y}+L_{f}\dot{\varphi }}{{\dot{x}}^{2}} & 0 & \frac{L_{f}}{\dot{x}} & 0 & 0\\ \frac{1}{\dot{x}} & -\frac{\dot{y}-L_{r}\dot{\varphi }}{{\dot{x}}^{2}} & 0 & -\frac{L_{r}}{\dot{x}} & 0 & 0 \end{array}\right] \end{array}\right.$$
where $\alpha ={\left[\alpha_{f},\alpha_{r}\right]}^{T}$ is the tire corner angle matrix, $F={\left[-1,0\right]}^{T}$ is the direct transfer matrix, and E is the output matrix.

Based on the above objective function and constraints, the optimization problem of the controller can be described as:(24)$$\left\{ \begin{array}{l} \min_{\Delta U,\varepsilon }\sum_{i=1}^{N_{p}}{\left\|\Delta \eta \left(t+i|t\right)\right\|}_{Q}^{2}+\sum_{i=1}^{N_{c}-1}{\left\|\Delta u\left(t+i|t\right)\right\|}_{R}^{2}+\rho \varepsilon^{2}\\ \Delta U_{\min}\leq \Delta U\leq \Delta U_{\max}\\ \Delta U_{\min}\leq A\Delta U+U_{f}\leq \Delta U_{\max}\\ \alpha_{\min}\leq \Delta \alpha \leq \Delta \alpha_{\max}\\ \varepsilon \geq 0 \end{array}\right.$$

Solving the above formula can get the incremental sequence of control input in each control time domain:(25)$$\Delta U^{*}\left(k\right)=\left[\Delta u^{*}\left(k/k\right),\Delta u^{*}\left(k+1/k\right),\cdots ,\Delta u^{*}\left(k+N-1/k\right)\right]$$

Apply the first element of the incremental sequence to the controller as the actual input increment:(26)$$u\left(k/k\right)=u\left(k-1\right)+\Delta u^{*}\left(k/k\right)$$

Repeating the above process, the optimal control input to the front wheel angle can be obtained.

#### 3.2.3. Multi-Sensor Information Data Fusion and Fault Signal Isolation

If multi-sensors are used to measure one vehicle motion parameter, we can fuse the outputs of all sensor system using the weight assignment method [28], which can be written as follows:(27)$$O=WMI=\left[w_{1},w_{2},\cdots ,w_{n}\right]diag\left[m_{1},m_{2},\cdots ,m_{n}\right]{\left[i_{1},i_{2},\cdots ,i_{n}\right]}^{T}$$
where *O* is the result of data fusion; $W=\left[w_{1},w_{2},\cdots ,w_{n}\right]$ is the weight matrix; $I={\left[i_{1},i_{2},\cdots ,i_{n}\right]}^{T}$ is outputs of each sensor; *n* is the number of sensors; and M is fault detection matrix.

The principle of the method can be written as follows:(28)$$\left\{ \begin{array}{l} w_{j}=1/\left[\sigma_{j}^{2}\sum_{i=1}^{n}\left(1/\sigma_{i}^{2}\right)\right]\\ \sum_{i=1}^{n}w_{j}=1 \end{array}\right.$$
where $\sigma_{i}$ and $\sigma_{j}$ are the output error dispersion of the *i*-th and *j*-th sensors; i,j=1,2,…n.

The true values of vehicle motion states cannot be obtained, and the traditional methods determine the average value of the different sensors outputs as the true value. However, these methods are not suitable in our situation, since different sensors in different conditions can give a deliberately low accuracy or even failure; due to this reason, the average value may have a large difference with the true value, which is extremely harmful to vehicle safety.

If at moment *k*, the sensor *j* gives the measured value $T_{j}\left(k\right)$, then:(29)$$\left\{ \begin{array}{l} \Delta T_{j}\left(k\right)=T_{j}\left(K\right)-{\hat{T}}_{j}\left(k\right)\\ \Delta {\bar{T}}_{j}=\frac{1}{N}\sum_{k=1}^{N}\Delta T_{j}\left(k\right)\\ \sigma_{j}\left(k\right)=\frac{1}{N}{\sum_{k=1}^{N}\left[\Delta T_{j}\left(k\right)-\Delta {\bar{T}}_{j}\right]}^{2} \end{array}\right.\;\;k=1,2,\cdots ,N$$
where $∆T_{j}\left(k\right)$ is the measurement error of the *j*-th sensor at time *k*; $∆{\bar{T}}_{j}$ is the average value of the *j*-th sensor at moment *k*; $\sigma_{j}\left(k\right)$ is the variance of the output error of the *j*-th sensor at time *k*; $\hat{T_{j}}\left(k\right)$ is prognostic assessment obtained using the Kalman filter; *N* is the number measurements from each sensor.

Since the following filtering process includes the isolation of unreliable data sources and error correction, we can approximately consider the estimated information as the true value.

### 3.3. Fault Signal Detector Design

Figure 3 shows the block diagram of proposed faults signals detector.

Each filter in this detector is standard, we take the Sub_Kalman_Filter_1 for GPS signal channel as an example. The state vector and measurement vector can be written as:(30)$$\left\{ \begin{array}{l} X_{GPS}=\left[\Delta \dot{x},\Delta \dot{y},\Delta \varphi ,\Delta \dot{\varphi },\Delta X,\Delta Y\right]\\ Y_{GPS}=\Delta \varphi \end{array}\right.$$

The state-space and measurement-space equations of Kalman Filter from moment *k* − 1 to moment *k* can be written as:(31)$$\left\{ \begin{array}{l} X_{GPS,k}=F_{GPS,k}X_{GPS,k-1}+w_{GPS,k}\\ Y_{GPS,k}=H_{GPS,k}X_{GPS,k}+v_{GPS,k} \end{array}\right.$$
where $F_{GPS,k}$ is the state transition matrix, $W_{GPS,k}$ is the process noise, $H_{GPS,k}$ is the measurement transition matrix, and $v_{GPS,k}$ is the measurement noise.

The equations of Kalman Filter can be written as:(32)$$\left\{ \begin{array}{l} {\hat{X}}_{GPS,k|k-1}=F_{GPS,k}{\hat{X}}_{GPS,k-1|k-1}\\ P_{GPS,k|k-1}=F_{GPS,k}P_{GPS,k-1|k-1}F_{GPS,k}^{T}+Q_{GPS,k}\\ K_{GPS,k}=P_{GPS,k|k-1}H_{GPS,k}^{T}{\left(H_{GPS,k}P_{GPS,k|k-1}+R_{GPS,k}^{T}\right)}^{-1}\\ {\hat{X}}_{GPS,k|k}={\hat{X}}_{GPS,k|k-1}+K_{GPS,k}\left(Y_{GPS,k}-H_{GPS,k}{\hat{X}}_{GPS,k|k-1}\right)\\ P_{GPS,k|k}=\left(I-K_{GPS,k}H_{GPS,k}\right)P_{GPS,k|k-1} \end{array}\right.$$
where $P_{GPS,k|k}$ is covariance matrix, $K_{GPS,k}$ is the gain matrix, $Q_{GPS,k}$ is the process noise covariance matrix, and $R_{GPS,k}$ is the measurement noise covariance matrix.

The equation of state propagator can be written as:(33)$$\left\{ \begin{array}{l} {\hat{X}}_{i,k|k-1}=F_{i,k}{\hat{X}}_{i,k-1|k-1}\\ P_{i,k|k-1}=F_{i,k}P_{i,k-1|k-1}F_{i,k}^{T}+Q_{i,k} \end{array}\right.,\;\;i=1,2$$

The $\chi^{2}$ test method is widely used to detect faults in stochastic dynamic systems based on correspondence between the observed and reference signals [29]. This method can be divided into three types: $\chi^{2}$ test for residual error; $\chi^{2}$ test for state with a single state propagator; $\chi^{2}$ test for state with double state propagators. These methods have their own advantages and disadvantages:

(1) If the test statistics are calculated using the residual error, it is almost impossible to detect the fault in the state transfer process, although the fault of the sensors can be easily detected;

(2) If the test statistics are calculated using the state vector, it is possible to detect the fault in the state transfer process and to evaluate the fault of the sensor indirectly. If only one state propagator is used for correction of the state prediction error, the error accumulates and may diverge with time increases;

(3) If two state propagators are used and alternately reset the outputs, then accumulation of errors can be avoided. However, for this method, if *M* sub-filters are used, then *2M* state propagators should be used, which not only complicates the structure of the algorithm but also affects the speed of calculations.

It can be seen that these methods have their advantages and disadvantages. In this work, a novel robust fault detector is proposed with a structure that simultaneously implements a $\chi^{2}$ test for residual error and a $\chi^{2}$ test for state. Double state propagators are used only for the main Kalman filter (see Figure 3), which allows to simplify the structure of the algorithm and increase the speed of calculations. As an example, Figure 4 shows the diagram of the Fault_Dignal_Detector_1.

As long as one test channel detected a fault, it can be considered as fault signal. In addition, since the state vectors of all filters are the same, we use only two state propagators. Derive the state $\chi^{2}$ test with two state propagators and define error state vectors as:(34)$$\left\{ \begin{array}{l} e_{i,k}=X_{k}-{\hat{X}}_{i,k}\\ e_{GPS,k}=X_{k}-{\hat{X}}_{GPS,k} \end{array}\right.$$
where $X_{k}$ is true state vector, ${\hat{X}}_{GPS,k}$ is estimation from Sub_Kalman_Filter_1 for GPS signal channel, and ${\hat{X}}_{i,k}$ is estimation error from state propagator *i*. In this paper, we consider the following variable to detect fault signal from Sub_Kalman_Filter_1 for GPS signal channel:(35)$$\beta_{GPS,k}=e_{i,k}-e_{GPS,k}={\hat{X}}_{GPS,k}-{\hat{X}}_{i,k}$$

The variance of this variable can be written as:(36)$$T_{GPS,k}=E\left\{\beta_{GPS,k}\beta_{GPS,k}^{T}\right\}=E\left\{e_{GPS,k}e_{GPS,k}^{T}-e_{GPS,k}e_{i,k}^{T}-e_{i,k}e_{GPS,k}^{T}+e_{i,k}e_{i,k}^{T}\right\}=P_{GPS,k}-P_{GPS\_i,k}-P_{i\_GPS,k}+P_{i,k}$$
where $P_{GPS\_i,k}$ and $P_{i\_GPS,k}$ is the cross-covariance.

We set the same initial conditions for the Sub_Kalman_Filter_1 for GPS signal channel and the state propagator *i*, then we can obtain $P_{GPS\_i,k}=P_{i\_GPS,k}=P_{i,k}$, therefore, the variance can be written as:(37)$$T_{GPS,k}=P_{GPS,k}-P_{i,k}$$

Define the fault detection function:(38)$$\lambda_{GPS,k}=\beta_{GPS}^{T}T_{GPS,k}^{-1}\beta_{GPS}^{}$$

The fault decision criteria can be written as:(39)$$\left\{ \begin{array}{l} \lambda_{GPS,k}\geq \varepsilon_{\beta },\;fault\\ \lambda_{GPS,k}<\varepsilon_{\beta },\;no\;faul \end{array}\right.$$
where the threshold $\varepsilon_{\beta }$ is determined by the function of false alarm rate based on statistical results.

In this paper, we use the state $\chi^{2}$ test with two state propagators. The working principle can be described as follows: during period $t_{k}$ a fault occurs and the switch $K_{1}$ is located at position “ $L_{1}$”, switch $K_{2}$ is located at position “ $L_{2}$”, the output of State_Propagator_1 is uncorrected due to this fault, but the output of State_Propagator_2 is obtained using the previous correct state, which can be used for fault correction. During period $t_{k+1}$, errors in State_Propagator_1 are corrected using the outputs of Kalman filter. After a time period ∆t, the $K_{1}$ switch is located at position “$L_{2}$”, the switch $K_{2}$ is located at position “$L_{1}$”, and the State_Propagator_2 is used to correct the fault.

Test $\chi^{2}$ residual error of Sub_Kalman_Filter_1 can be written as:(40)$$d_{GPS,k}=\Delta \varphi_{GPS,k}-\Delta {\hat{\varphi }}_{i,k}$$

Covariance residual error:(41)$$S_{GPS,k}=H_{GPS,k}P_{GPS,k}H_{GPS,k}^{T}+R_{GPS,k}$$

Define the fault detection function:(42)$$\gamma_{GPS,k}=d_{GPS,k}^{T}S_{GPS,k}^{-1}d_{GPS,k}^{}$$

Fault decision criteria can be written as:(43)$$\left\{ \begin{array}{l} \gamma_{GPS,k}\geq \varepsilon_{d},\;fault\\ \gamma_{GPS,k}<\varepsilon_{d},\;no\;fault \end{array}\right.$$
where the threshold $\varepsilon_{d}$ is determined by the function of false alarm rate based on statistical results.

If the state $\chi^{2}$ test or the residual error $\chi^{2}$ test detected a fault, then $m_{g}=0$, otherwise $m_{g}=1$.

## 4. Simulation Experiment Verification and Discussion

### 4.1. Working Conditions Description

In order to verify the feasibility and effectiveness of the proposed method, the Driving Scenario Designer was used to build a simulated driving environment with two straight lanes, and the reference path and yaw angle were collected, as shown in Figure 5a. The system model was built in the Matlab/Simulink environment.

In Figure 5a, the simulation vehicle starts from the right lane and then changes lanes to the left. From Figure 5b, we can observe that the simulation vehicle is initially located at *x* = −10, *y* = 0, and then changes lanes at *x* = 0, *y* = 0. The lane changing process is completed at *x* = 80 m, and the position of the center of mass on the *Y*-axis reaches *y* = 4 m.

Figure 6a shows the values of vehicle yaw angle obtained by different sensors in the simulation process. Figure 6b shows the reference yaw angle of the vehicle and the yaw angle after data fusion.

The horizontal axis represents time, and the vertical axis represents the values of yaw angle. It can be seen that from moment 0s to 3s, the measured values of Vision, GPS, and LIDAR systems are similar. In the simulation process, we assume that the GPS signal has interfered from 3 to 6 s, and the obtained yaw angle value has a large deviation. After 6s, the GPS signal returns to normal.

It can be seen that the fused yaw angle is always less than the value of the reference yaw angle before 2.3s and the fused yaw angle is greater than the value of the reference yaw angle between 2.3s to 5s. According to the trend of the two curves in the image, we know that the path tracking control performance of straight line is significantly better than the curve line during the process of lane change.

The type of simulated autonomous vehicle used in our paper is passenger car. The simulation environment and initial simulation condition settings are shown in Table 1:

### 4.2. Effectiveness of the Proposed Method

The yaw angle errors are shown in Figure 7.

It can be seen that except GPS, the deviation between other measurement data and reference value is extremely small, and their absolute values are not more than 0.01 rad. However, when it comes to GPS data, the deviation is about 0.7rad at 3s, which is unacceptable considering the maxim yaw angle about 0.09 rad.

Figure 8 shows the value of simultaneous interpreting of three different sensors based on Chi-square test.

Figure 8a is the value obtained by state chi-square test, and Figure 8b is the value obtained by residuals error chi-square test. In Figure 8a, the calculated values λ of state chi-square test of Vision, GPS, and LIDAR channels are very small, which are always below the order of magnitude $10^{-7},$ when all sensor systems work without fault. In contrast, when the fault of GPS signal occurs, the calculated values of state chi-square test of GPS channel sharply increase at 3s, reaching more than the order of magnitude 5× $10^{-6}$. In Figure 8b, the calculated values γ of state residuals test of Vision, GPS, and LIDAR channels are generally not exceed 2 × $10^{-8},$ when all sensor systems work without fault. However, in the period of 3s-5s, when the GPS signal has fault, the calculated values γ of residuals error chi-square test of GPS signal channel is much higher, and even reaches more than 5 × $10^{-7}$ at 3 s.

Figure 9a shows the lateral position changes of the vehicle obtained by Vision, GPS, and LIDAR sensor systems. Figure 9b shows the reference lateral position of the vehicle and the lateral position after data fusion.

It can be seen from Figure 9 that the lateral position obtained by GPS signal has obvious deviation in the time period of 3-6s, and the lateral position obtained by multi-sensor data fusion method is basically consistent with the reference lateral position. However, a small deviation between the reference and fused lateral position still exists due to the performance of path tracking controller.

Figure 10 shows the deviation of the measured lateral position by different sensor systems.

It can be seen from Figure 10 that the measurement deviations of vision and LIDAR have remained at a very small range, while the measurement error of GPS data gradually increases from about 1.5m to more than 3m in the period of 3-6s due to sensor failure. Considering that the lane width is only 4m, the GPS sensor fault is unacceptable and needs to be isolated.

Figure 11 shows the value of the fault detection function calculated by chi-square test for different sensor systems. Figure 11a shows the value obtained by state chi-square test and Figure 11b shows the value obtained by residuals chi-square test.

In Figure 11a, it can be seen that the calculated values of λ state chi-square test of vision and LIDAR channels are very small, basically below 2 × $10^{-5},$ occasionally exceeding the order of magnitude $10^{-5}$, but they are still extremely small. Comparatively, the data of GPS channel reaches the order magnitude of $10^{-3}$ at 3 s, or even more than $10^{-2}$ at 6 s, which is much larger than that of vision and LIDAR channels. In Figure 11b, the calculated value γ of vision channel reaches about 1 × $10^{-6}$ at the initial stage but always below the order magnitude of $10^{-6}$ in the following time. The calculated value γ of GPS data is extremely large from 3 s to 6 s, reaching the order magnitude of $10^{-3}$ due to the GPS sensor failure. The calculated values γ of LIDAR data fluctuate greatly, but they are all below 1 × $10^{-5}$, which is also a reasonable interference situation.

From the above description and analysis of Figure 7 to Figure 11, it can be seen that the proposed method can detect the fault signal robustly when the sensor failure occurs.

Figure 12 shows the simulation results of path tracking control with and without fault isolation. The three curves in Figure 12a represent the reference lateral position, after fault isolation and before fault isolation. Figure 12b shows the lateral position error before and after fault isolation in the simulation process. From Figure 12a,b, it can be seen that without fault isolation, it is almost impossible to realize the path tracking control due to sensor faults. After fault isolation, the deviation between the actual lateral position and the reference lateral position is maintained within a very small range, which validates the effectiveness of the proposed fault-tolerant MPC algorithm for path tracking of autonomous vehicle.

Figure 12c,d show the yaw angle obtained before and after fault isolation and the yaw angle error with respect to the reference values. From Figure 12c,d, it can be seen that with fault isolation, actual yaw angle of vehicle almost coincides with the reference yaw angle, while without fault isolation the yaw angle error is extremely large after 1s. From Figure 12d, the yaw angle error without fault isolation can reach—0.5 rad to 1.1 rad, while the reference value of yaw angle is always below 0.1rad, which means yaw angle error without fault isolation may reach 5–10 times more than the reference, and vehicle cannot track the reference path under this condition, which may lead to serious consequences. On the contrary, after fault isolation processing, the yaw angle error is always below 0.02 rad, which is extremely small for path tracking control.

### 4.3. Discussion of the Background and Outcomes of Our Work

With the continuous increasing requirements for the safety and environmental adaptability of autonomous vehicles, the on-board environmental perception systems have become more and more complex, and the types and numbers of sensors have also increased. The risk of sensor failures is also increased, which has a serious impact on vehicle safety. Therefore, the detection of faulty sensor signals and fault-tolerant control mechanism is very important to autonomous driving safety. In our work, we established a single-track 3 DOF vehicle dynamics model and based on this model, a fault-tolerant model predictive control method was developed. Our motivation for designing this algorithm is to enable the autonomous vehicle to effectively detect and isolate the fault signal and to perform robust longitudinal path tracking motion control when the sensor failure occurs.

In order to verify the effectiveness of the method proposed in this paper, we set up a single lane changing path tracking control condition in Driving Scenario Designer. The vehicle motion state information, such as the lateral position and yaw angle, can be obtained by the GPS integrated navigation system, LIDAR perception system, and visual perception system. We assume that there is interference in the simulation environment, which causes the GPS signal to be temporarily lost. The description and analysis of simulation results show that the proposed fault-tolerant MPC algorithm can effectively detect the fault signal when the sensor failure occurs. The value of fault detection functions of fault yaw angle signal and fault lateral position signal is 10 and 100 times, respectively, more than that of normal signals. This means that we can easily detect the fault signal by setting the appropriate threshold and generate the corresponding fault detection matrix. The fault isolation is accomplished in the process of data fusion, which is one of the innovations of our work. With the fault signal isolation, the path tracking control performance of autonomous vehicle can be significantly improved, further confirming the effectiveness and robustness of the proposed algorithm in this work.

## 5. Conclusions

In this paper, a novel and robust fault-tolerant model predictive control algorithm was proposed, which can be used for robust vehicle lateral motion control in case of sensor failures. First, by constructing the objective function and considering dynamic constraints, the linear time-varying model predictive control algorithm for path tracking control of vehicle was designed. Second, an improved weighted data fusion algorithm was proposed for multi-sensor information fusion and fault signal isolation. Then, based on Chi-square detectors and Kalman filters, we designed a novel fault signal detection algorithm, which can produce the fault detection matrix used in data fusion algorithm for fault signal isolation. Finally, a lane changing path tracking control simulation was carried out for validation of the effectiveness and correctness of the proposed algorithm. The simulation results show that the proposed algorithm can efficiently detect the fault signal. After the fault signal isolation, the reference path can be effectively tracked by using the proposed fault-tolerant MPC algorithm. In further studies, this method can be combined with reinforcement learning to improve fault detection and isolation performance. We will explore the effectiveness of the proposed method for fault detection in electronic fuel injection system, automatic steering control system, suspension systems, etc. Meanwhile, we will extend our fault-tolerant model predictive control algorithm into the fault diagnosis fields that are out of the real driving environment, for instance, fault diagnosis in complex driving scenarios and strong environmental noise conditions. In addition, considering the longitudinal speed changes and the adaptive MPC algorithm can be used to improve the path tracking performance of motion controller. It is worth noting that we approximately use the fused multi-sensor data as the real motion states of vehicle, which means that when faults occur to most or all sensors, our proposed method is invalid. In addition, the robustness is weak by using the thresholds to detect the fault signal in our method. For solving these two problems, using the convolutional neural network to automatically extract the features of the fault detection function and detect the sensor fault could be a reliable solution.

## Figures and Tables

**Figure 1 sensors-20-04245-f001:**
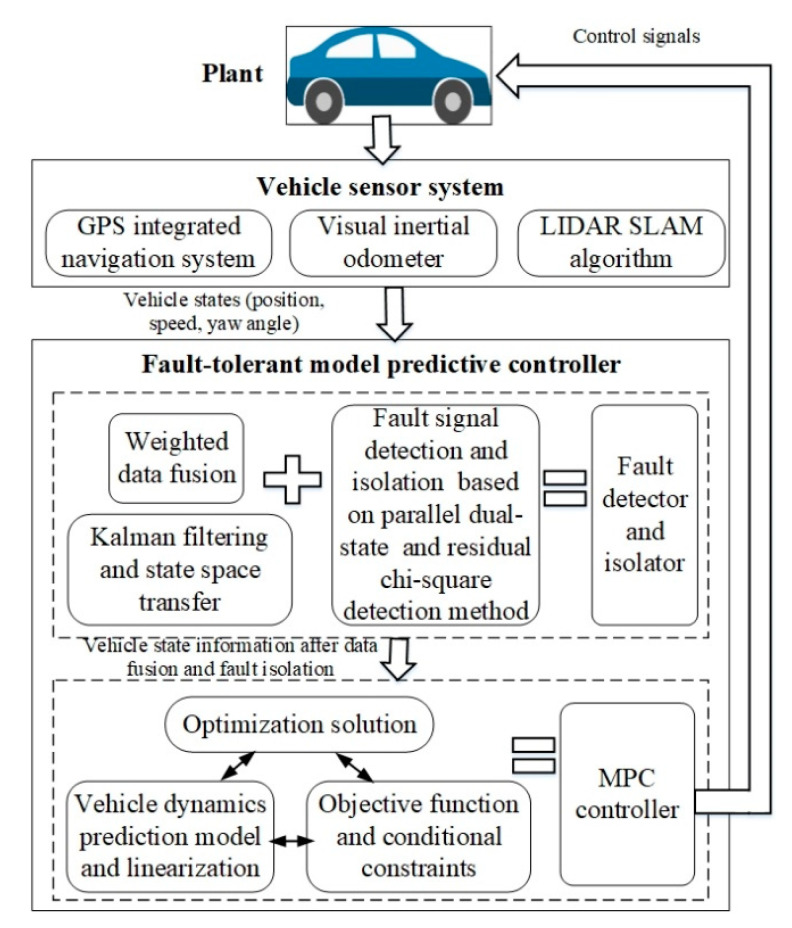
The flowchart of the fault-tolerant model predictive control algorithm.

**Figure 2 sensors-20-04245-f002:**
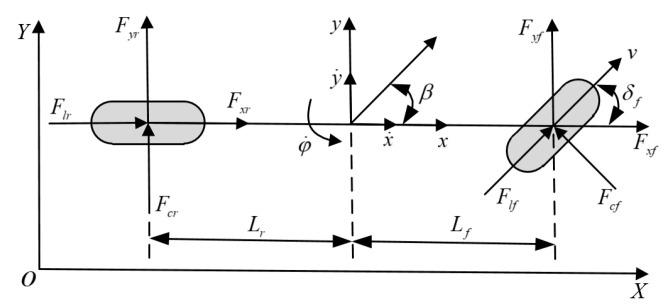
Schematic of the single-track vehicle model.

**Figure 3 sensors-20-04245-f003:**
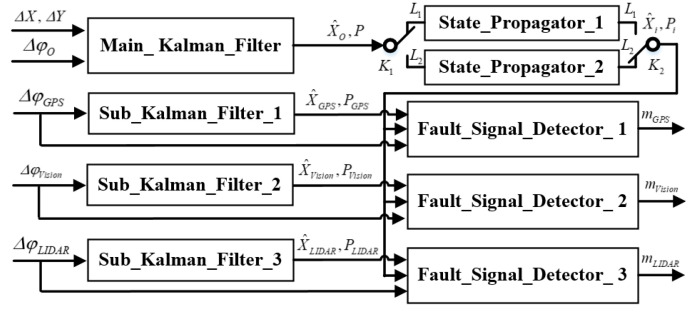
The block diagram of proposed faults signals detector.

**Figure 4 sensors-20-04245-f004:**
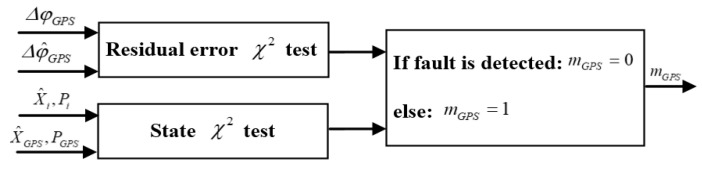
The diagram of the Fault_Signal_Detector_1.

**Figure 5 sensors-20-04245-f005:**
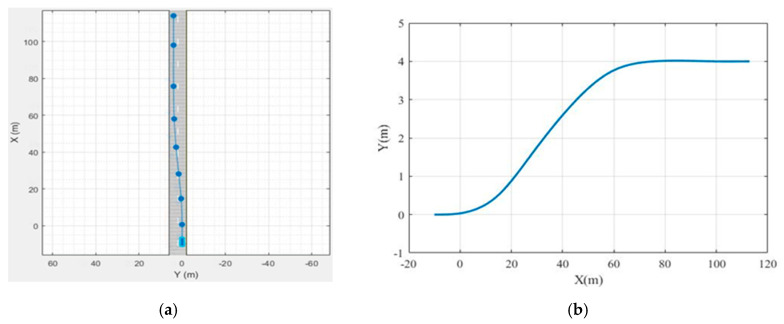
Driving scenario and reference path: (**a**) driving scenario; (**b**) reference path.

**Figure 6 sensors-20-04245-f006:**
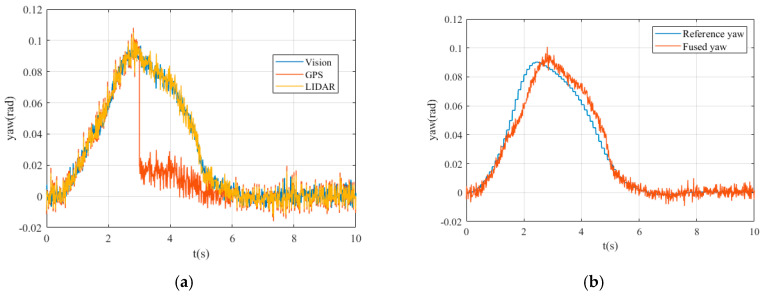
The yaw angle: (**a**) yaw angle detected using different sensors; (**b**) the reference and fused yaw angle.

**Figure 7 sensors-20-04245-f007:**
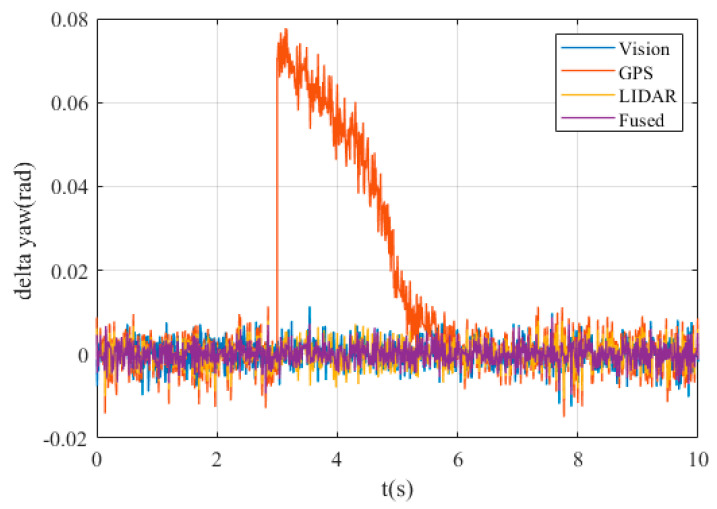
The yaw angle errors.

**Figure 8 sensors-20-04245-f008:**
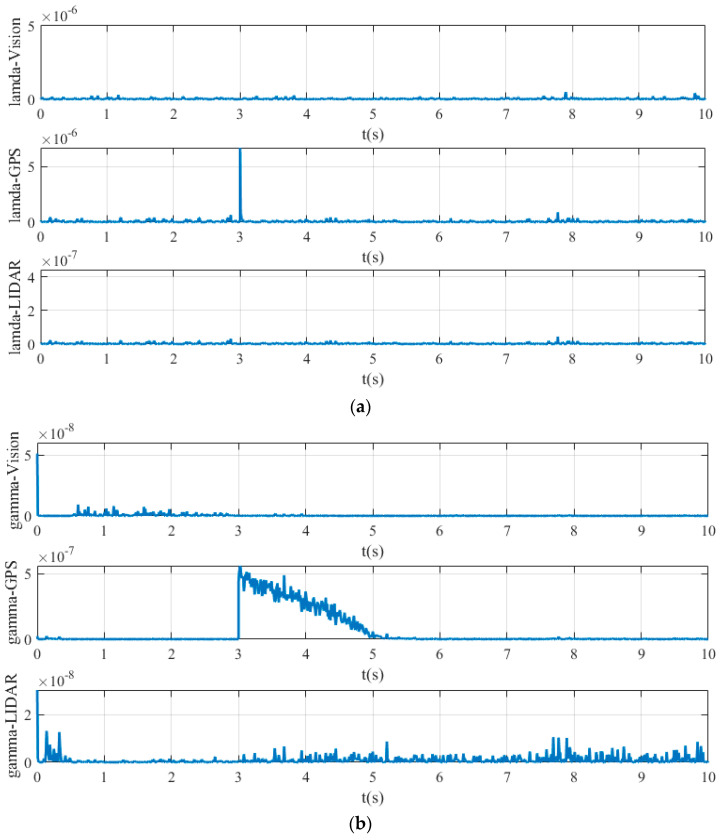
Yaw test statistic of: (**a**) state Chi-square test; (**b**) residuals Chi-square test.

**Figure 9 sensors-20-04245-f009:**
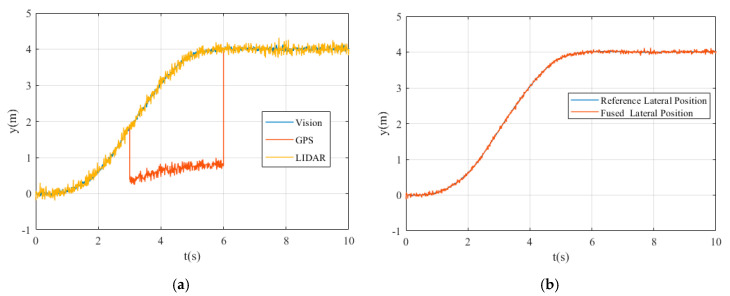
The lateral position: (**a**) lateral position detected using different sensors; (**b**) the reference and fused lateral position.

**Figure 10 sensors-20-04245-f010:**
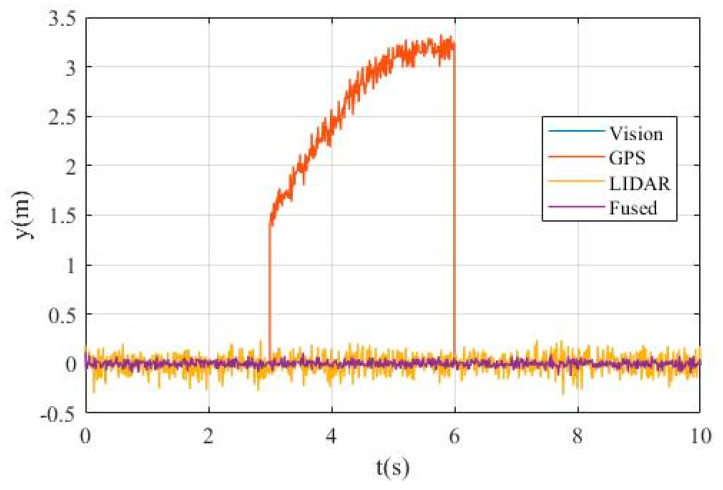
The lateral position error.

**Figure 11 sensors-20-04245-f011:**
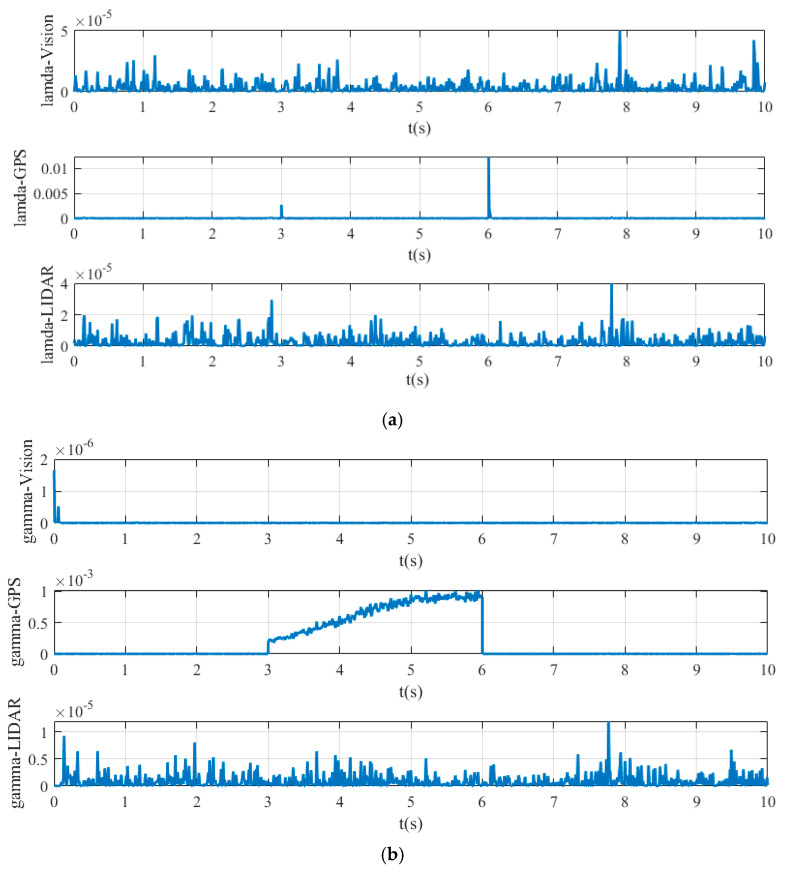
Lateral position test statistic of: (**a**) state Chi-square test; (**b**) residuals Chi-square test.

**Figure 12 sensors-20-04245-f012:**
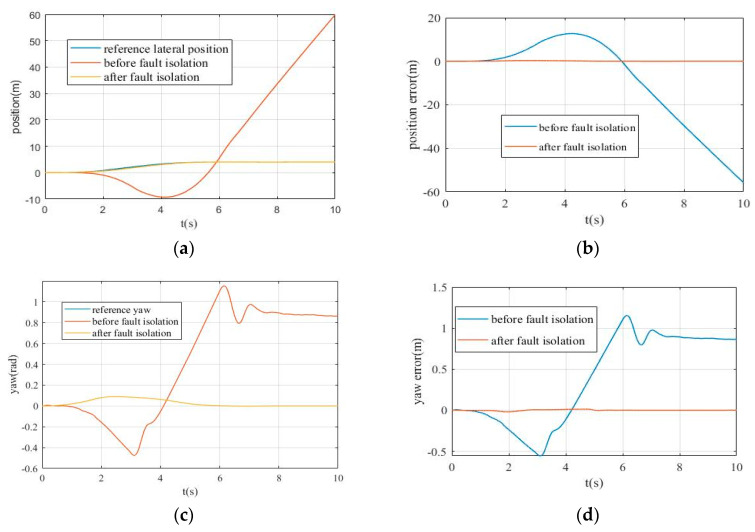
The path tracking control simulation results: (**a**) the lateral position changes; (**b**) the lateral position error before and after fault isolation; (**c**) the yaw angle changes; (**d**) the yaw angle error before and after fault isolation.

**Table 1 sensors-20-04245-t001:** The simulation parameters.

Parameters	Value	Dimension
Number of lanes:	2	
Lane width:	4	m
Lane length	127	m
Mass of car:	1575	kg
Moment of inertia	2875	kg.m^2^
Length of car	4.7	m
Width of car	1.8	m
Height of car	1.4	m
Front overhang	0.9	
Rear overhang	1	
Longitudinal speed	15	km/h
Sampling time interval	0.01	s
The process noise covariance matrix Q	diag [0.001,0.001,0.001,.0.002]	
The measurement noise R	0.001

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
