# Peer review of "Fault-Tolerant Model Predictive Control Algorithm for Path Tracking of Autonomous Vehicle"

_sensors, 2020, doi:10.3390/s20154245_

Round 1
Reviewer 1 Report
1-On line 21, what do you mean by "were fused"?!
2-How tire forces have been approximated by a linear function?
3-What type of car is simulated in Table 1? It is needed to add the type of simulated vehicle in the Table 1.
Author Response
Point 1: 1-On line 21, what do you mean by "were fused"?! 

Response 1: Dear professor, first of all, thank you very much for your comments!
The vehicle motion states information can usually be obtained by GPS integrated navigation system, visual odometer and LIDAR SLAM. These sensors and algorithms together form an on-board perception system. The effective working conditions of each type of sensors are not the same. For example, the camera cannot work normally when the illumination conditions are insufficient. The detection accuracy of LIDAR sensors are easily affected by bad weather such as smoke, rain and snow. The GPS signal is easily blocked in the environment with tall trees, urban buildings and tunnels, which can cause signal interruption. Therefore, the sensors of autonomous vehicle may be influenced by illumination, signal outage, poor weather conditions, magnetic interference and other environmental circumstances. In addition, the sensors age and inherent errors may also result in sensor failure and affecting data availability. The method of sensor redundancy and data fusion can be used for avoidance sensor fault. If multi-sensors are used to measure one vehicle motion parameter, for instance the yaw angle, we can fuse the yaw angle outputs of all sensor systems by using the weight assignment method to enhance the reliability of measurements.
Point 2: How tire forces have been approximated by a linear function?
Response 2: Thanks for your comments.
According to the Semi-Empirical Tire-Model [23], we know that the tire dynamics, including the tire longitudinal force, the tire lateral force and the tire aligning torque, have obvious nonlinear characteristics, but the simulation results in [24-25] show that the tire forces can be approximated by a linear equation, when the longitudinal slip rate and the tire variation angle changes in a small range. In addition, there are a large number of trigonometric functions in the vehicle dynamics model. Since each angle involved in the dynamics model is in a small angle interval, each trigonometric function can satisfy the following approximate conditions: , , . After introducing the corner stiffness, corner angle, longitudinal stiffness and slip rate, the tire force of the vehicle can be expressed as:
Please see the attachment
Point 3: What type of car is simulated in Table 1? It is needed to add the type of simulated vehicle in the Table 1.
Response 3: Thanks again for your comments.
The type of simulated autonomous vehicle in our paper is passenger car. We have added the type of simulated vehicle in our paper.

Reviewer 2 Report
This paper presents the development of an innovative algorithm which designed for path tracking control of an autonomous vehicle in case of a fault signal occurrence. Also, a simulation was implemented for validating the developed algorithm, which is very important. The methodological approach is suitable and well-explained. The results are properly substantiated, shedding light into the primary aim of the study. Most of the sections of the manuscript are properly structured and explained.
The following comments are suggested to be taken into consideration at the revised version of this paper.
The introduction section is appropriate, all arguments are justified based on literature review. The methodological approach (which can be found in sections 2 and 3) is properly defined, and every single step has its mathematical equation and its explanation which leads to a better understanding of the study. At the section on “simulation experiment verification”, all results concerning the simulation model are also properly substantiated. The simulation section is appropriate to verify the effectiveness of the proposed methodological approach and subsequently to address adequate the aim of the study.
However, a discussion section is required, in order to shed light on the outcomes and their background based on the research results. Furthermore, at the discussion section it is highly recommended to compare the results with those of the literature review, whenever this is possible.
At the conclusions section, a brief paragraph with the most important outcomes of the study would be beneficial. In addition, in the existing manuscript there is only one suggestion for further research, hence it is recommended to be further expanded with more arguments. Moreover, the study limitations should also be added.
Also, it is recommended to increase the quality of figures, since some words are not properly legible.
A final language and grammar improvement is suggested by a native speaker.
Author Response
Point 1: However, a discussion section is required, in order to shed light on the outcomes and their background based on the research results. Furthermore, at the discussion section it is highly recommended to compare the results with those of the literature review, whenever this is possible.
Response 1: Dear professor, first of all, thank you very much for your comments! We all agree your arguments and we added a discussion section in our paper:
4.3 Discussion of the background and outcomes of our work
With the continuous increasing requirements for the safety and environmental adaptability of autonomous vehicles, the on-board environmental perception systems have become more and more complex, and the types and numbers of sensors have also increased. The risk of sensor failures are also increased, which has a serious impact on the vehicle safety. Therefore, the detection of faulty sensor signals and fault-tolerant control mechanism are very important to autonomous driving safety. In our work, we established a single-track 3 DOF vehicle dynamics model, and based on this model, a fault-tolerant model predictive control method was developed. Our motivation for designing this algorithm is to enable the autonomous vehicle to effectively detect and isolate the fault signal and to perform robust longitudinal path tracking motion control, when the sensor failure occurs.
In order to verify the effectiveness of the method proposed in this article, we set up a single lane changing path tracking control condition in matlab Driving Scenario Designer. The vehicle motion state information such as the lateral position and yaw angle can be obtained by the GPS integrated navigation system, LIDAR perception system and visual perception system. We assume that there is interference in the simulation environment, which causes the GPS signal to be temporarily lost. The description and analysis of simulation results show that the proposed fault-tolerant MPC algorithm can effectively detect the fault signal when the sensor failure occurs. The value of fault detection functions of fault yaw angle signal and fault lateral position signal are 10 and 100 times, respectively, more than that of normal signals. This means that we can easily detect the fault signal by setting the appropriate threshold and generate the corresponding fault detection matrix. The fault isolation is accomplished in the process of data fusion, which is one of the innovations of our work. With the fault signal isolation, the path tracking control performance of autonomous vehicle can be significantly improved, further confirming the effectiveness and robustness of the proposed algorithm in this work.
According to the literatures we have reviewed, there are very limited research works related to the lateral path tracking fault-tolerant control of autonomous vehicle. Reproducing fault detection methods for other objects (such as, electronic fuel injection system, automatic steering control system, and suspension systems et. al.) is a very difficult task. In our subsequent research, we will compare with some data-driven fault tolerant control methods based on deep convolutional neural networks.
Point 2: At the conclusions section, a brief paragraph with the most important outcomes of the study would be beneficial. In addition, in the existing manuscript there is only one suggestion for further research, hence it is recommended to be further expanded with more arguments. Moreover, the study limitations should also be added.
Response 2: Thanks for your comments.
The most important outcomes of the study is that we developed a novel and robust fault-tolerant model predictive control algorithm for robust vehicle lateral motion control in case of sensor failures. The lane change path tracking simulation results show the effectiveness and correctness of the proposed algorithm. We added several suggestions for further research at the conclusions section. In addition, we concluded the study limitations and provided possible solution. Thanks to your comments, we changed the conclusions section to:
In this paper, a novel and robust fault-tolerant model predictive control algorithm was proposed, which can be used for robust vehicle lateral motion control in case of sensor failures. First, by constructing the objective function and considering dynamic constraints, the linear time-varying model predictive control algorithm for path tracking control of vehicle was designed. Second, an improved weighted data fusion algorithm was proposed for multi-sensor information fusion and fault signal isolation. Then, based on Chi-square detectors and Kalman filters, we designed a novel fault signal detection algorithm, which can produce the fault detection matrix used in data fusion algorithm for fault signal isolation. Finally, a lane change path tracking simulation was carried out for validation the effectiveness and correctness of the proposed algorithm. The simulation results show that the proposed algorithm can efficiently detect the fault signal. After the fault signal isolation, the reference path can be effectively tracked with the proposed fault-tolerant MPC algorithm. In the further studies, this method can be combined with reinforcement learning to improve the fault detection and isolation performance. We will explore the effectiveness of the proposed method for fault detection in electronic fuel injection system, automatic steering control system, suspension systems, etc. Meanwhile, we will extend our fault-tolerant model predictive control algorithm into the fault diagnosis fields that are out of the real driving environment, for instance fault diagnosis in complex driving scenarios and strong environmental noise conditions. In addition, considering the longitudinal speed change of the vehicle, and the adaptive MPC algorithm can be used to improve the path tracking performance of motion controller. It is worth noting that we approximately use the fused multi-sensor data as the real motion states of vehicle, which means that when faults occur to most or all sensors, our proposed method is invalid. In addition, the robustness is weak by using the thresholds to detect the fault signal in our method. For these two problems, using the convolutional neural network to automatically extract the features of the fault detection function and detect the sensor fault could be a reliable solution.
Point 3: Also, it is recommended to increase the quality of figures, since some words are not properly legible.
Response 3: Thanks again for your comments.
We updated the images in our paper and the quality of figures should have been improved.
Point 4: A final language and grammar improvement is suggested by a native speaker.
Response 4: Thanks again for your comments.
Each author of this work has read the paper carefully again and corrected some of the grammatical errors, but since English is not our native language, there may be some sentences that still can be improved. We will strengthen our writing skills in English in the future.
Finally, thank you very much for your comments, which are very valuable for the improvement of our paper.
